# Acute Foggy Corneal Epithelial Disease: Seeking Clinical Features and Risk Factors

**DOI:** 10.3390/jcm11175092

**Published:** 2022-08-30

**Authors:** Fei Li, Ruibo Yang, Liu Yang, Yuanyuan Qi, Chen Zhang, Yue Huang, Shaozhen Zhao

**Affiliations:** Tianjin Key Laboratory of Retinal Functions and Diseases, Tianjin Branch of National Clinical Research Center for Ocular Disease, Eye Institute and School of Optometry, Tianjin Medical University Eye Hospital, Tianjin 300384, China

**Keywords:** corneal epithelial disease, foggy vision, epithelial cells edema, female

## Abstract

(1) Purpose: Here, we describe the clinical characteristics and predisposing factors of acute foggy corneal epithelial disease, a rare disease newly found during COVID-19 pandemic; (2) Methods: In this single-arm, ambispective case series study, ten patients with acute foggy corneal epithelial disease admitted between May 2020 and March 2021 were enrolled. Their detailed medical history and clinical and ophthalmic findings were recorded and analyzed; (3) Results: All the patients were female (100%), aged from 28 to 61 years (mean age of 40.4 ± 9.3 years). Seven cases (70%) had excessive eye use, and six cases (60%) had stayed up late and were overworked. Ten subjects (100%) presented with acute onset and a self-healing tendency. There was a mild-to-moderate decrease in the corrected visual acuity (0.35 ± 0.21 (LogMAR)). Slit-lamp examination showed diffuse dust-like opacity and edema in the epithelial layer of the cornea. By in vivo confocal microscope, epithelial cells presented characteristically a “relief-like” appearance. Anterior segment optical coherence tomography examination revealed that the mean epithelial thickness was increased (69.25 ± 4.31 μm, *p* < 0.01); (4) Conclusions: Acute foggy corneal epithelial disease is a rare disease in clinic, which tends to occur in young and middle-aged females. The typical clinical symptom is sudden foggy vision, which occurs repeatedly and can be relieved without treatment. Sex, an abnormal menstrual cycle, overuse of the eyes, fatigue and pressure might be risk factors. Changes in lifestyle and eye use habit during the COVID-19 pandemic may have possibly contributed to this disease incidence.

## 1. Introduction

The epithelium is the outermost layer of the cornea, with a barrier function that makes it crucial for resisting the invasion of various toxins and pathogenic microorganisms [1,2]. The corneal epithelium can proliferate and renew itself, and repair rapidly after injury. This rapid renewal and the tight junctions between the cells are the main barrier to prevent the cornea from being invaded and damaged by harmful substances and pathogens [3,4,5,6]. Importantly, any adverse factors in the surrounding environment can affect the function of the corneal epithelium and induce damage [7,8,9,10]. There are many causes of corneal epithelial diseases, which are mainly divided into primary and secondary lesions. In the previous two years, since the COVID-19 outbreak, we have observed a special class of corneal epithelial disease, which still cannot be clearly classified according to the existing known corneal diseases, and there is very limited relevant literature and few case reports. We provisionally termed it as acute foggy corneal epithelial disease. In this study, we reported a case series of ten patients with this disorder. The etiology, clinical features and manifestations of in vivo confocal microscope are preliminarily described and investigated, with the intention of drawing the attention of ophthalmologists to this disease and facilitating further diagnosis and treatment.

## 2. Materials and Methods

In this single-arm, ambispective observational study, ten patients with acute foggy corneal epithelial disease who were admitted to the Tianjin Medical University Eye Hospital between April 2020 and March 2021 were enrolled consecutively. A questionnaire was provided to obtain disease history, symptoms, systemic history, medication history and other possible predisposing factors in detail. Routine ophthalmologic examination, including visual acuity, intraocular pressure (IOP) and slit-lamp examination were performed. The anterior segment photography, anterior segment optical coherence tomography (OCT) (RTVue; Optovue, Inc., Fremont, CA, USA) and in vivo confocal microscopy (HRT-III RCM, Heidelberg Engineering GmbH, Heidelberg, Germany) were performed during disease onset and recovery. The information from the questionnaire and the examination results were analyzed and summarized. This study followed the tenets of the Declaration of Helsinki and was approved by the Tianjin Medical University Eye Hospital Ethical Committee.

Data were expressed as the mean ± standard deviation (SD). The continuous data were analyzed firstly by the Kolmogorov–Smirnov test for normality test. Statistical comparison of corneal epithelial thickness at onset and the recovery stage was performed using paired *t*-tests (data follow a normal distribution). SPSS software version 23.0 (New York, NY, USA) was used for statistical analysis. Statistical significance was considered to be *p* < 0.05.

## 3. Results

### 3.1. Patients’ Demography and Possible Predisposing Factors

Ten patients were included, all of whom were females (10/10, 100%), aged from 28 to 61 years, with mean age of 40.4 ± 9.3 years. Seven cases (7/10, 70%) had excessive eye use, that is, staying up late or long use of electronic devices (daily ≥ 8 h or continuous use ≥ 4 h) before onset. Three patients had poor sleep quality, and one elderly subject among them had a short sleep time (3–4 h sleep/day). Six patients (6/10, 60%) had high work pressure and fatigue. We further investigated the menstrual period and hormone-related situations. An abnormal menstrual cycle and estrogen levels were found in six patients (6/10, 60%). One had a history of pregnancy before onset, one was perimenopausal and one was menopausal. Additionally, all patients had varying degrees of dry eye history. Three patients had a history of laser-assisted in situ keratomileusis (LASIK) surgery, and the first episode of acute foggy corneal epithelial disease appeared, respectively, 2 months, 1 year and 10 years after LASIK surgery. Three patients had a history of allergy, while the remainder had no obvious systemic or surgical history. All the patients and their families had no COVID-19 infection history or COVID-19 vaccination. The predisposing factors are presented Table 1.

### 3.2. Clinical Features during the Onset Stage

The disease had an acute onset and both eyes were involved in all cases. The typical symptoms of the onset included mild eye redness, foggy vision, mild-to-moderate vision decrease, photophobia, and were occasionally accompanied by tears. The mean corrected distance visual acuity was 0.35 ± 0.21 (LogMAR). The symptoms could be spontaneously alleviated or relieved (10/10, 100%), but recurred frequently. The symptoms lasted for several hours (10/10, 100%) to several days (3/10, 100%). Among the ten patients, six cases (6/10, 60%) developed symptoms within 1–2 h after morning rise, and the other four patients (4/10, 40%) had irregular onset time. Slit-lamp examination showed mild hyperemia of the conjunctiva, no follicles or papillae, diffuse dust-like opacity and edema in the epithelial layer of cornea, stained with fluorescein (Figure 1). No abnormalities of the corneal stroma and endothelium were observed. Additionally, no intraocular abnormalities were observed. The IOP was normal.

### 3.3. Auxiliary Examinations

Typical findings of in vivo confocal microscope examination included epithelial cells swelling, particularly basal cells with a “relief-like” high reflection, a low reflective cytomembrane of basal cells with an unclear boundary, no significant decrease in the density of sub-basal nerve plexus, an absence of inflammatory cells and activated Langerhans cells in all corneal layers, all anterior stromal keratocytes being activated or normal, and no significant abnormality appearance of the endothelial cells (Figure 2). The epithelial basal cells displayed high reflectivity in the cell body and low reflectivity at the cell boundary, which was reverse of that of normal basal cells (low reflectivity in the cell body and high reflectivity at the cell boundary). This change gives the basal cells a three-dimensional appearance; therefore, it is called a relief-like appearance.

Anterior segment OCT examination revealed a bright reflection on the corneal epithelium. The mean epithelial thickness was increasing to 69.25 ± 4.31 μm (*p* < 0.01) compared with the recovery stage (Figure 3).

### 3.4. In the Recovery Stage

Corrected distance visual acuity was completely restored in all patients (−0.02 ± 0.04 (LogMAR)). The cornea was transparent under the slit lamp. In vivo confocal microscopy showed slightly high reflection of epithelial basal cells or normal cells of the cornea. Anterior segment OCT examination revealed that the epithelial thickness had returned to normal, with a mean thickness of 51.25 ± 2.82 μm. The comparison of clinical manifestations between the onset and recovery stages are presented in Table 2.

## 4. Discussion

The corneal epithelium consists of nonkeratinized lamellar squamous epithelial cells that maintain the transparency of the cornea through their inherent immune protection and barrier functions [12]. The epithelial cells are connected by tight junctions, which are closely combined to prevent the invasion of most microorganisms and the tears from entering the stroma layer, thus maintaining the cornea in a relatively dehydrated state [5,13,14]. In addition, there are microvilli on the outer cytomembrane of the epithelial cells [1,15]. These microvilli extend into the tear membrane to absorb the tear and prevent the epithelial cells from drying. The physiological balance of corneal epithelial cell proliferation and differentiation helps to maintain the state of dehydration, transparency and normal visual function [16].

It has been reported that environmental stress, including ultraviolet (UV) radiation, hyperosmotic pressure, hypoxia and infection, could cause corneal epithelial cells damage, cell volume change, mesenchymal structure disorder and inflammatory cell infiltration [17,18]. UVB delayed the self-renewal and wound healing of corneal epithelial cells by activating a variety of signaling pathways, including Kv channels, mitogen-activated protein kinase signaling pathways, and K^+^ channels [18,19,20]. 

Clinically, mild-to-moderate corneal epithelial edema can cause a cloudy and microcystic appearance, resulting in vision loss and glare. In severe cases, it can also cause pain, photophobia and subepithelial bullae. These changes are associated with degeneration of edematous epithelial basal cells, which is caused by accumulation of fluid in the epithelium and between the epithelium cells to form the cysts and bullae.

The main clinical manifestations in this case series were sudden bilateral foggy vision and corneal epithelial edema with both normal endothelium structure and IOP, with the symptoms lasting for several hours with spontaneous remission and could recur frequently. No inflammatory evidence was found under confocal microscopy in vivo. Because this type of disorder was rarely reported in previous literature works, we provisionally named it acute foggy corneal epithelial disease. According to the clinical features, we made a differential diagnosis from a similar disease: glaucopsia.

Glaucopsia is a transient visual impairment resulting from exposure to vapors from certain industrial chemicals, particularly amines, aliphatic compounds, and aliphatic rings [21,22,23]. The cases have been mainly associated with workers at chemical factories. After an incubation period of 30 min to several hours of exposure, blurred vision usually develops, and objects take on a blue-gray appearance [21]. This visual impairment is sometimes likened to seeing through smoke, and it can cause a halo around bright objects and even photophobia in severe cases [22,23]. The main changes that can be observed under slit-lamp microscope are corneal edema and vesicular accumulation of intracellular fluid under the corneal epithelium [24]. Increased corneal thickness can be measured with a thickness gauge [25]. The symptoms subside and vision returns to normal within hours of cessation of exposure to the vapors. These ocular effects do not cause permanent damage to the eyes [26]. A concentration–effect relationship has been established. There are correlations between vapor concentration, exposure duration, degree of corneal edema, and subjective symptoms. In terms of the pathological mechanism, Ballantyne et al. suggested that the diffusion of amine vapor to the corneal surface caused epithelial edema and subepithelial microcapsules, with the accumulation of water on the corneal surface leading to light scattering and foggy vision [22].

In the present case series, the symptoms and signs of these patients were very similar to the corneal epithelial damage caused by amine compounds, but none of the patients worked in an amine-exposed environment, and their living and working environment remained unchanged. However, it was notable that due to the outbreak of the COVID-19 pandemic in the past two years, the frequency of using disinfectants, including alcohol- and chlorine-containing preparations, increased significantly both at home and in the workplace. In our study, two patients were frequently exposed to disinfectants. Although no reliable evidence has been obtained, it cannot be ruled out that the exposure to certain substances in the atmosphere caused the damage to the corneal epithelium.

The first patient we observed with this disorder was a young person who underwent refractive surgery. We initially thought that the disease might be one of the complications of this surgery, thus we needed to make a differential diagnosis from central toxic keratopathy (CTK).

CTK is initially described as a rare complication after LASIK surgery, mainly occurring within 9 days after refractive surgery, and patients may experience blurred vision, pain, photophobia, floaters, and foreign body sensation [27,28,29]. CTK presents as a central or peri-central amorphous corneal opacity with corneal striation, stroma thinning, and hyperopia shift. It is non-inflammatory and normally resolves within 2–18 months [30,31,32]. In addition to the cases of refractive surgery, CTK has been reported to be associated with the use of contact lenses with or without recent mechanical debridement, idiopathic and selective laser trabeculoplasty or topical anesthesia. Therefore, some scholars suggested that CTK might be an independent disease of refractive surgery [30].

CTK usually occurred during the early stage after refractive surgery; however, the onset of the three cases in our study were all later than 9 days. The main lesion was in the corneal epithelium rather than the stroma layer. There was no thinning of the corneal stroma or a hyperopia shift. CTK normally persists for several months before resolving spontaneously. Therefore, the diagnosis of CTK can be excluded by comparing the disease onset time after operation, lesion location, duration of disease resolution, and causes of the disease in patients without a refractive surgery history.

Additionally, a sudden foggy opacity of the cornea, which is observed by slit lamp, can readily be misdiagnosed as an acute attack of glaucoma. Therefore, attention should be paid to the IOP and the corneal endothelium and pupil, which were normal in acute foggy corneal epithelial disease.

Because all the patients in this study were female, we proposed that estrogen or the menstrual cycle-induced hormone level changes contributed to the development of corneal epithelial disease.

In recent years, sex differences in the anatomy, physiology and pathophysiology of human corneas have been confirmed. The corneal epithelial thickness is significantly greater in men than in women [33,34]. Similar to adults, the corneal epithelium is thicker in male children than in female children [35]. Epithelial regeneration and corneal wound repair are significantly delayed in female patients, suggesting that the ability of ocular surface wound healing is poor in females [36]. In addition, the corneal sensitivity, wetting time, epithelial mitosis rate and corneal graft survival rate were also affected by sex [37].

Suzuki et al. showed that there were sex-related differences in gene expression in human corneal epithelial cells. Sex significantly affects the expression of more than 600 genes in human corneal epithelial cells [37]. These genes are involved in a wide range of biological processes, molecular functions and cellular components. For example, the gene expression associated with DNA replication and cell migration is also significantly higher in male corneal epithelial cells. In females, there are other highly expressed genes, including transglutaminase 1. This enzyme catalyzes covalent cross-linking of proteins, and its expression normally increases in dry eye and corneal keratosis, possibly contributing to the greater prevalence of dry eye in females [38].

Sex differences in gene expression in these corneal epithelial cells may be mediated by estrogen and androgen secretion levels [37]. Estrogen is associated with a significant increase in the signs and symptoms of dry eye and induces the expression of pro-inflammatory genes in corneal epithelial cells and meibomian glands. Significant changes in estrogen levels during the menstrual cycle, pregnancy, menopause, and hormone replacement therapy can also induce changes in corneal hydration, curvature and sensitivity and in vision [37].

Furthermore, it has been reported that females are prone to be more sensitive to stress [39]. Stress was significantly associated with menstrual cycle irregularity [40]. Gonadal hormones are involved in regulating the stress response [41].

During the period of the COVID-19 pandemic, the pressures of life and work have increased, and the long-term economic and social consequences brought on by the pandemic have had a huge negative impact on mental health. According to the survey data published in the Lancet, the COVID-19 pandemic increased the prevalence of major depressive disorders by 27.6% and anxiety disorders by 25.6% in 2020 globally, particularly in females, who are more likely to be affected by the social and economic consequences of the pandemic than males [42].

In the present study, all the cases were female, and six out of the ten cases had an irregular menstrual cycle and abnormal hormone levels. One had a history of pregnancy before onset, one was perimenopausal and one was menopausal. Stress, staying up late and a lack of sleep could also cause changes in endocrine levels. Abnormal hormone levels might be one of the predisposing factors of this disease. The mechanism of sex differences in the prevalence of this disease remains unclear and needs further investigation.

It is interesting that the disease was discovered during the COVID-19 pandemic. The general lifestyle has changed significantly after the outbreak, including home quarantine, online working and conferences, increased time indoors, decreased outdoor activities, and long-term use of video display terminals. All these changes caused excessive use of the eyes, asthenopia, dry eyes and the homeostasis disruption of the ocular surface microenvironment. Moreover, while the use of a facemask is effective in preventing the transmission of COVID-19, prolonged or incorrect mask use can result in dry eyes or a potential risk on ocular surface health [43,44]. Incorrectly fitted facemasks reverse the normal airflow direction during nasal breathing and fail to protect the eyes from airflow during inspiration and expiration. This leads to the ocular surface being exposed to mechanical desiccation and nasopharyngeal pathogens [45].

In the present study, all patients had an acute and reversible foggy epithelial edema with normal IOP and endothelial structure. The results of fluorescein sodium staining of the corneal epithelium suggested that the function and integrity of the corneal epithelial barrier were disrupted. The imitation of the study is that corneal epithelial tissue or cells hardly can be obtained for intensive pathological or molecular mechanism studies, because the corneal epithelium was intact during the onset stage. We propose that it may be a form of stress reaction of the ocular surface, which leads to a decrease in tear film stability or even an abnormal tear composition, which, in turn, leads to the impairment of epithelial barrier function and even intercellular or intracellular edema. Therefore, the underlying pathogenesis of this disease remains unclear and needs to be further investigated.

## 5. Conclusions

Acute foggy corneal epithelial disease is a relatively rare disease; it tends to occur in young and middle-aged female. An abnormal menstrual cycle, fatigue and overuse of the eyes could be the main risk factors, but multiple factors may be involved. The onset is acute, with both eyes involved. The typical clinical symptoms are foggy vision and photophobia, which occur repeatedly and can be resolved without treatment. By in vivo confocal microscope, the epithelial basal cells display edema and are highly reflective, presenting a characteristic “relief-like” appearance. Inflammatory cells and activated Langerhans cells are both absent in all corneal layers. The cases are still being followed up, with no visual loss or other eye damage found to date. Changes in lifestyle and eye use habits during the period of the COVID-19 pandemic may have contributed to this disease incidence. The predisposing factors and pathogenesis of the disease remain unclear, and its clinical diagnosis and treatment need to be further standardized.

## Figures and Tables

**Figure 1 jcm-11-05092-f001:**
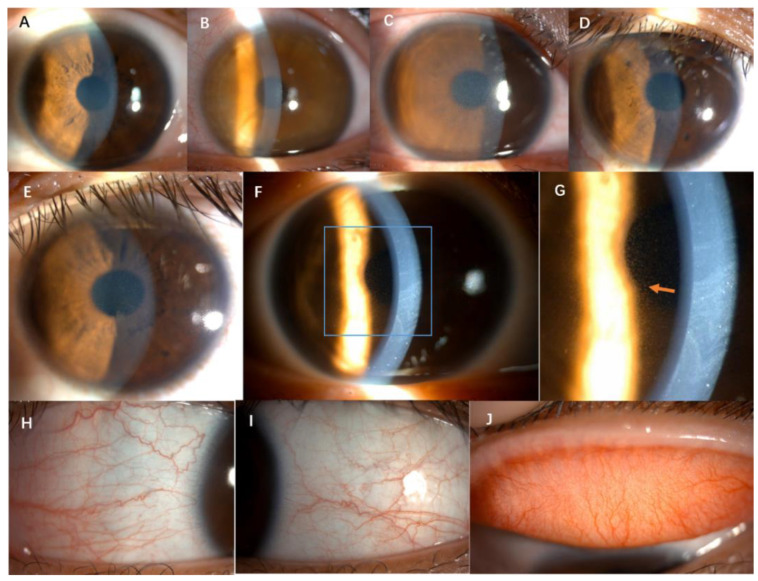
Anterior segment slit-lamp photography of six cases. (**A**–**F**), corneal diffuse dust-like opacity and edema; (**G**), the arrow shows diffuse dust-like opacity in the cornea epithelium by retro illumination. (**H**–**J**), mild hyperemia of the conjunctiva, no follicles or papillae (1/10 case).

**Figure 2 jcm-11-05092-f002:**
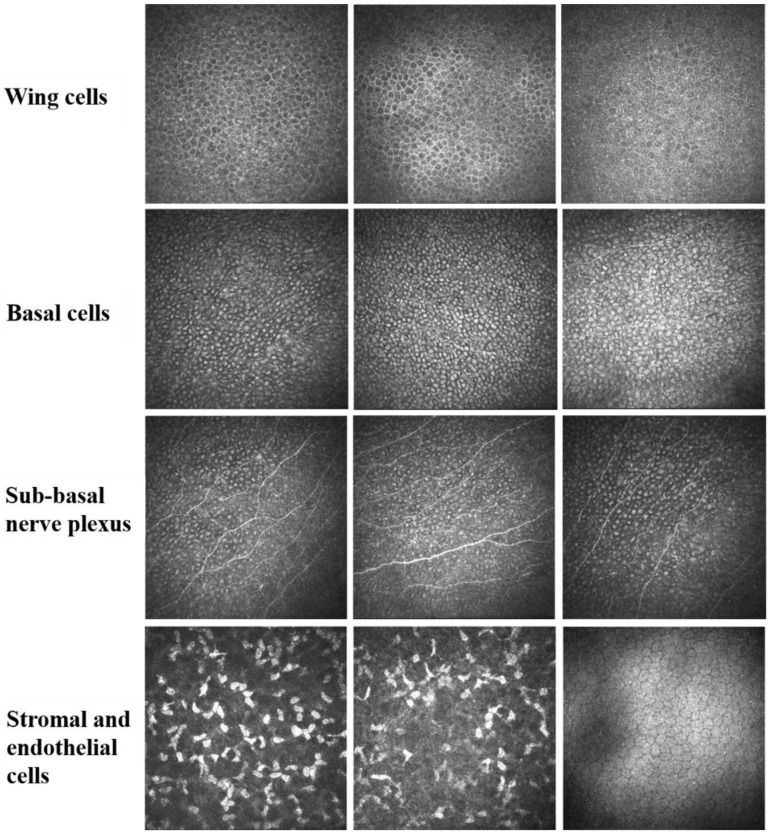
In vivo confocal microscope shows that epithelial cells are edematous with high reflectivity inside the basal cells, but low reflectivity at the cell boundary. This change shows a special stereoscopic appearance, presenting a “relief-like” manifestation. The density of sub-basal nerve plexus displays no significant decrease. Inflammatory cells and activated Langerhans cells are rarely seen. Anterior stromal keratocytes are activated or normal. There is no significant abnormality in the endothelial cells.

**Figure 3 jcm-11-05092-f003:**
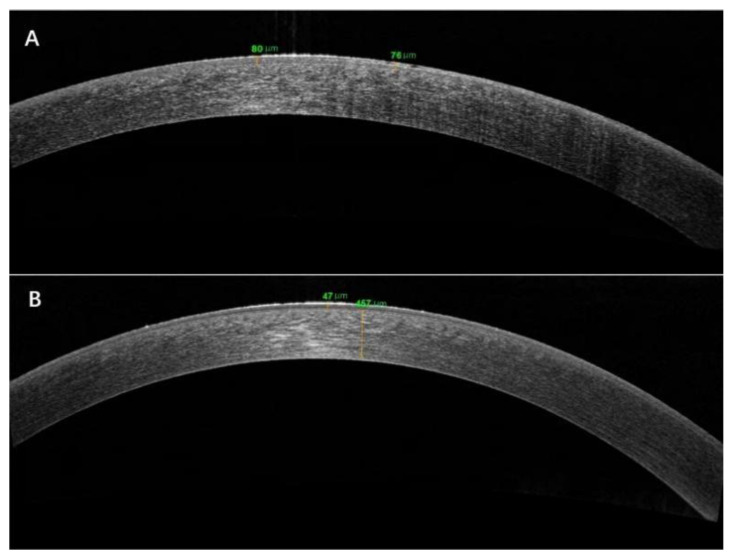
Anterior segment OCT examination revealed epithelial thickness in the onset (**A**) and recovery stages (**B**).

**Table 1 jcm-11-05092-t001:** Clinical Characteristics of Study Population.

Variable	Patient Data
Age, mean (SD), y	28–61, 40.4 ± 9.3
Female	10/10 (100)
Eyestrain (daily ≥ 8 h or continuous use ≥ 4 h)	7/10 (70)
Psychosocial stress	6/10 (60)
Overwork	5/10 (50)
Poor sleep quality	3/10 (30)
Staying up late	6/10 (60)
Refractive laser correction surgery	3/10 (30)
Contact lenses	2/10 (20)
Vaccination (cervical cancer vaccine)	1/10 (10)
Vaccination (COVID-19 vaccine)	0/0 (0)
Use of new cosmetics	2/10 (20)
Use of disinfectant	1/10 (10)
Irregular menstrual cycle and menopause	6/10 (60)
History of dry eye	10/10 (100)
History of allergy	3/10 (30)
History of COVID-19 infection (including family members)	0/0 (0)
History of medication (therapy for irregular menstruation)	2/10 (20)

Unless otherwise indicated, data are expressed as number/total number (percentage) of patients.

**Table 2 jcm-11-05092-t002:** Clinical Manifestations.

	The Onset Stage	The Recovery Stage
corrected visual acuity (LogMAR), mean (SD)	0.35 ± 0.21	0.0
IOP	normal	normal
conjunctival hyperemia	1 ^a^	0 ^a^
cornea opacity	0.5–2 ^b^	0 ^b^
epithelium	epithelial cells were edema, basal cells showed a “relief-like” appearance	slightly high reflection of epithelial basal cells or normal cells
stroma	anterior stromal keratocytes were activated or normal	normal
endothelium	normal	normal
inflammation in cornea	none	none
anterior chamber reaction	none	none

^a^ For conjunctival hyperemia scoring, the images were graded from 0 to 3. The criteria were: 0: absence of hyperemia; 1: mild hyperemia of the conjunctival vessels; 2: diffuse hyperemia of the conjunctival vessels; and 3: severe hyperemia of the conjunctival vessels [11] ^b^ Corneal opacity was graded from 0 to 3 as follows: 0: clear cornea; 0.5: very mild opacity seen only under dissection microscope; 1: mild opacity with visibility of iris details; 2: moderate opacity with partial masking of iris and visible pupil margin; 3: severe opacity with invisibility of iris and pupil details [11].

## Data Availability

The datasets generated and analyzed during the current study are available from the corresponding author on reasonable request.

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
