# Peer review of "Acute Foggy Corneal Epithelial Disease: Seeking Clinical Features and Risk Factors"

_jcm, 2022, doi:10.3390/jcm11175092_

Round 1
Reviewer 1 Report
The thought behind this paper is good; however, my main concern with this manuscript is the admission by the authors that there are a couple of limitations, which I believe renders the conclusion a little less than convincing. The authors should ensure that all abbreviations or acronyms used in the manuscript for the first time are written in full as well as ensure that sentences with minor punctuation and/or grammatical errors are revised. The authors were able to identify the possible predisposing factors and pathological changes in the corneal epithelial layer, but they did not provide any discussion of the molecular pathogenesis of the condition.
Lines 17 - 18: The authors should reword this sentence. This could be an incomplete sentence.
Line 44: Render “of ophthalmologist on this disease” as “of ophthalmologist to this disease”
Line 67: The authors should reword this sentence. This could be an incomplete sentence.
Line 71: What impact did this varying degree of dry eye have on the results of this study?
Lines 84 - 86: The authors should reword this sentence to render it more comprehensible.
Line 108: Provide an alternate rendering for this sentence. Probably render “edema” as “edematous”
Line 109: Elaborate on this “relief-like” appearance. This should apply to lines 19 and 99.
Line 134: Render “functions11.” as “functions [11]”.
Discussion section: The authors should elaborate on the molecular pathogenesis of this acute foggy corneal epithelial disease.
Lines 182 – 183: The authors should reword this sentence to render it more comprehensible.
Line 183: CTK - The authors should ensure that all abbreviations or acronyms used in the manuscript for the first time are written in full. This should apply to the rest of the manuscript.
Line 232: Render “stress38.” as “stress [38].
Line 247 – 248: Provide an alternate rendering for “state”
Line 258: Render “imitation” as “limitation”
Author Response
Dear Reviewer,
Thank you very much for your comments and professional advice. These opinions help to improve academic rigor of our article. Based on your suggestion and request, we have made corrected modification on the revised manuscript. Meanwhile, the manuscript had be reviewed and edited by language services. We hope that our work can be improved again. Furthermore, we would like to show the details as follows:
- Lines 17 - 18: The authors should reword this sentence. This could be an incomplete sentence.
Response:This sentence has been reworded to “There was a mild to moderate decrease in the corrected visual acuity (0.35±0.21 (LogMAR)).”
- Line 44: Render “of ophthalmologist on this disease” as “of ophthalmologist to this disease”
Response:“of ophthalmologist on this disease” has been modified to “of ophthalmologist to this disease”.
- Line 67: The authors should reword this sentence. This could be an incomplete sentence.
Response:This sentence has been reworded to “Three patients had poor sleep quality, and one elderly subject among them had a short sleep time (3-4hrs sleep/day).”
- Line 71: What impact did this varying degree of dry eye have on the results of this study?
Response:In our study, dry eye was provided by patients as medical history and medication history (artificial tears) data. In the onset period, patients who generally had some common symptoms and signs, including photophobia, tearing and corneal epithelial edema, cannot be well coordinated with a series of dry eye examinations, so the severity of dry eye cannot be accurately assessed. In addition, considering artificial tears had been applied during the process of the disease, the ocular surface situation at the follow up was not representative of the initial situation. Therefore, the initial severity of dry eye was not evaluated in this study.
- Lines 84 - 86: The authors should reword this sentence to render it more comprehensible.
Response:These sentences have been reworded to “The symptoms could be spontaneously alleviated or relieved (10/10, 100%), but recurred frequently. The symptoms lasted for several hours (10/10, 100%) to several days (3/10, 100%). Among the ten patients, six cases (6/10, 60%) developed symptoms within 1-2 hours after morning rise, and the other four patients (4/10, 40%) had irregular onset time.”
- Line 108: Provide an alternate rendering for this sentence. Probably render “edema” as “edematous”
Response:“edema” has been modified to “edematous”
- Line 109: Elaborate on this “relief-like” appearance. This should apply to lines 19 and 99.
Response:The “relief-like” appearance has been described in detail on lines 99 and 109. Because of the word limit of the abstract, the content of lines 19 has not been changed.
Line 99: The epithelial basal cells displayed high reflectivity in the cell body and low reflectivity at the cell boundary, which was reverse of that of normal basal cells (low reflectivity in the cell body and high reflectivity at the cell boundary). This change gives the basal cells a three-dimensional appearance, therefore, it is called relief-like appearance.
Line 109: In vivo confocal microscope shows that epithelial cells are edematous with high reflectivity inside the basal cells, but low reflectivity at the cell boundary. This change shows a special stereoscopic appearance, presenting a “relief-like” manifestation.
- Line 134: Render “functions11.” as “functions [11]”.
Response:“functions11.” has been modified to “functions [11]”
- Discussion section: The authors should elaborate on the molecular pathogenesis of this acute foggy corneal epithelial disease.
Response:In our study, due to the fact that the corneal epithelium was intact during the onset stage, we could hardly obtain the corneal epithelial tissue or cells for intensive pathological or molecular mechanism studies. We also couldn’t find published articles addressing the molecular pathogenesis of this disease. We propose it may be a form of stress reaction of the ocular surface, which leads to a decrease in tear film stability or even an abnormal tear composition, which in turn leads to the impairment of epithelial barrier function and even intercellular or intracellular edema. Therefore, the underlying pathogenesis of this disease remains unclear. In the future, we hope to find appropriate methods to further explore the molecular pathogenesis.
At the end of the discussion section of the manuscript, we added a paragraph to explain the question.
- Lines 182 – 183: The authors should reword this sentence to render it more comprehensible.
Response: These sentences have been reworded to “We initially thought that the disease might be one of the complications of refractive surgery, thus we need to make differential diagnosis with central toxic keratopathy (CTK).”
- Line 183: CTK - The authors should ensure that all abbreviations or acronyms used in the manuscript for the first time are written in full. This should apply to the rest of the manuscript.
Response:Full words have been added in the manuscript for the first appearance.
- Line 232: Render “stress38.” as “stress [38].
Response: “stress38.” has been modified to “stress [38]”
- Line 247 – 248: Provide an alternate rendering for “state”
Response:“state” has been modified to “lifestyle”
- Line 258: Render “imitation” as “limitation”
Response:“imitation” has been modified to “limitation”
Thank you very much for your attention and time.
Best Wishes,
Fei Li

Reviewer 2 Report
This is an interesting and novel disease of the cornea with unknown etiology. The authors have addressed the possible differential diagnosis in the discusion section in detail. There are some points to address before the submission.
- Did the authors assess the normality test in the istatistic? If the data are not normally distributed, the statistical test should be repeated with the Wilcoxon test.
-The line 105, "The mean 104 epithelial thickness was increasing to 69.25±4.31um (p<0.0001) compared to recovery stage. " It is more appropriate to change the p value to p= 0.0001 or p< 0.01.
-The authors mentioned that the patients recovered without treatment, but there is no information about when patients recover time, in hours, days or weeks?
-the line 258, "The imitation of our study is that corneal epithelial tissue or cells can not be obtained" I think there is a typos error in this word, it should be "The limitation"
Author Response
Dear Reviewer,
Thank you very much for your comments and professional advice. These opinions help to improve academic rigor of our article. Based on your suggestion and request, we have made corrected modification on the revised manuscript. Meanwhile, the manuscript had be reviewed and edited by language services. We hope that our work can be improved again. Furthermore, we would like to show the details as follows:
- Did the authors assess the normality test in the istatistic? If the data are not normally distributed, the statistical test should be repeated with the Wilcoxon test.
Response:The data in the study were normally distributed. We have added a note in the statistics section.
The continuous data were analyzed firstly by Kolmogorov-Smirnov test for normality test. Statistical comparison of corneal epithelial thickness among onset and recovery stage was performed using paired t-tests (data follow a normal distribution).
- The line 105, "The mean 104 epithelial thickness was increasing to 69.25±4.31um (p<0.0001) compared to recovery stage. " It is more appropriate to change the p value to p= 0.0001 or p< 0.01.
Response:“p<0.0001” has been modified to “p<0.01”.
- The authors mentioned that the patients recovered without treatment, but there is no information about when patients recover time, in hours, days or weeks?
Response:In the part of 3.2, the recover situation has been descripted as “The symptoms lasted for several hours (10/10,100%) to several days (3/10, 30%)”.
- the line 258, "The imitation of our study is that corneal epithelial tissue or cells cannot be obtained" I think there is a typos error in this word, it should be "The limitation"
Response: “imitation” has been modified to “limitation”
Thank you very much for your attention and time.
Best Wishes,
Fei Li

Round 2
Reviewer 1 Report
The authors should cite the original source for the sentences in lines 112 – 116 and 162 – 167.